# Development and application of an online tool to quantify nitrogen removal associated with harvest of cultivated eastern oysters

**Julie M. Rose**[1]*, **Ryan Morse**[2,3], **Christopher Schillaci**[4]

**1** NOAA National Marine Fisheries Service, Northeast Fisheries Science Center, Milford Laboratory, Milford, Connecticut, United States of America, **2** NOAA Fisheries NEFSC Narragansett Laboratory, Narragansett, Rhode Island, United States of America, **3** f CASE Consultants International, Asheville, North Carolina, United States of America, **4** NOAA National Marine Fisheries Service, Greater Atlantic Regional Fisheries Office, Gloucester, Massachusetts, United States of America

* julie.rose@noaa.gov

## Abstract

Shellfish aquaculture can provide important ecosystem services to coastal communities, yet these benefits are not typically considered within the aquaculture permit review process. Resource managers have expressed interest in easy-to-use tools, based on robust science, that produce location and operation-appropriate values for beneficial services. These values need to be produced in a format that aligns with existing regulatory processes to facilitate seamless integration with permit review. The removal of excess nitrogen from coastal waters by shellfish farms is well documented in the literature and has been incorporated into nutrient management in the USA. Shellfish assimilate nitrogen into their tissue and shell as they grow, and this nitrogen is removed from the environment upon harvest. We have assembled a dataset of nitrogen concentration and morphometric measurements from farmed eastern oysters across the US Northeast, and adapted methodology used by existing nutrient management programs to quantify harvest-associated removal of nitrogen. Variability in oyster tissue and shell nutrient concentration was low within the dataset, and an assessment of farm location, ploidy, and three common cultivation practices (floating gear, bottom gear, no gear) suggested that a simple regression-based calculation could be applied across all farms within the region. We designed the new, publicly available online Aquaculture Nutrient Removal Calculator tool https://connect.fisheries.noaa.gov/ANRC/ based on this analysis, which uses inputs related to oyster size and harvest number to predict harvest-based nitrogen removal from an eastern oyster farm located within the geographic range of North Carolina to Maine, USA. The tool also produces a report that has been designed to integrate with the US Army Corps of Engineers public interest review process, and similar state-level permitting processes, and provides a succinct summary of the ecological services associated with nutrient removal in eutrophic locations, project-specific values, and citations supporting the calculation of those values.

**Data Availability Statement:** All data have been submitted to Zenodo and are freely available for

download at https://doi.org/10.5281/zenodo.11966672.

**Funding:** Funding for this work was provided to JMR and CS from the NOAA Fisheries Office of Aquaculture, https://www.fisheries.noaa.gov/about/office-aquaculture. There was no grant number associated with this funding. The funders played no role in the study design, data collection and analysis, decision to publish, or preparation of the manuscript.

**Competing interests:** The authors have declared that no competing interests exist.

## Introduction

Shellfish aquaculture can provide ecosystem services to society [1–3]. The most widely-recognized benefit is seafood production, but a broad range of other services have been documented in the literature. Shellfish farms can remove excess nutrients from the coastal environment [4–6], improve water clarity through filtration activities [7], and increase wave attenuation in nearshore waters [8, 9]. Shellfish aquaculture gear has been shown to provide habitat for wild fish and invertebrates (reviewed in [10]), and through provisioning of habitat to juvenile life stages, may result in the enhancement of production of species of commercial and recreational value [11]. Shellfish farms provide important cultural services to communities, through increased employment opportunities, increased social capital, an enhanced sense of place, and increased recreational opportunities [12].

Nitrogen removal by shellfish farms in eutrophic coastal environments is one ecosystem service that has received a great deal of attention by scientists and resource managers in the last decade [13–15]. Excess nitrogen has been linked to a variety of environmental problems in marine systems around the world, such as nuisance and harmful algal blooms, hypoxia, and loss of submerged aquatic vegetation that serves as nursery habitat for wild fish [16–18]. Nitrogen management programs have been implemented to reduce the symptoms of eutrophication in coastal environments across North America, Europe, Australia, and Asia [19]. Shellfish farms can remove excess nitrogen through three primary mechanisms: 1) assimilation of nitrogen into animal tissue and shell, which is removed from the environment when shellfish are harvested; 2) enhancement of naturally-occurring, microbially-mediated denitrification processes in marine sediments through the enrichment of carbon and nitrogen by shellfish biodeposits under aquaculture operations; and 3) long-term burial of shellfish biodeposits in marine sediments [20–22].

Of the three mechanisms for shellfish aquaculture nitrogen removal, assimilation of nitrogen into animal tissue and shell is best characterized in the scientific literature (as reviewed in [23]). Many of these studies have focused on blue mussels, *Mytilus edulis*, and eastern oysters, *Crassostrea virginica*, as these species are commonly farmed at high density in locations which also experience nutrient overenrichment (e.g. [24–26]). Nitrogen concentration (by percent of dry weight) has been observed to exhibit low spatial and temporal variation in oyster and mussel tissue and shell [23, 27]. The total weight of nitrogen removed by an individual animal is small, but the low variability in nitrogen concentration, and associated high degree of confidence in calculations of nitrogen removal by harvested shellfish, has made shellfish aquaculture an attractive potential contributor to nitrogen management in the United States and Europe [5, 6].

Shellfish aquaculture has been incorporated into nitrogen management programs in several locations in the United States. At the municipal scale, several towns on Cape Cod, Massachusetts, USA have integrated shellfish into their nitrogen removal programs. The Town of Mashpee has used a combination of hard clam stock enhancement and eastern oyster aquaculture to help meet water quality goals [28]. The Town of Falmouth has expanded eastern oyster aquaculture in local waters for nitrogen removal purposes [29, 30]. The only estuary-scale, multistate integration of shellfish aquaculture into nutrient management to date has occurred in Chesapeake Bay, USA [27]. The Chesapeake Bay Program approved the baywide use of eastern oyster aquaculture by local jurisdictions to meet nitrogen and phosphorus reduction goals in 2016. This approval was based on recommendations from an expert panel that synthesized available nutrient data for eastern oysters across the Northeast Region, and morphometrics data for eastern oysters within Chesapeake Bay [27].

While integration of nutrient removal services provided by shellfish aquaculture into nutrient management programs is growing in the US, recognition of this ecosystem service has not

been fully integrated within shellfish aquaculture permitting and management. One logical avenue for this integration is the US Army Corps of Engineers public interest review process (33 CFR § 320), as it requires consideration of the benefits of a proposal to be balanced against its reasonably foreseeable detriments during the permit application review process. The process specifically includes evaluation of a project's effects on water quality. Similar consideration of the benefits and adverse effects of proposed aquaculture projects to water quality is also required under regulations in a number of U.S. states. Despite the requirement to consider both adverse and beneficial effects, generally the review is limited to the consideration of adverse effects. We held a series of discussions with state and federal aquaculture permitting authorities in the states of Connecticut, New Jersey, and Massachusetts, and the North Atlantic Division of US Army Corps of Engineers regarding the status of, and potential barriers to, greater incorporation of environmental benefits in shellfish aquaculture permitting. Feedback from managers indicated that greater and more consistent integration could be attained if location and operational specific values were available that could easily be applied during the permit review process. In particular, managers indicated a need for simple tools, based on robust data, to aid in the consideration of environmental benefits in the permit review process.

In direct response to the feedback from aquaculture resource managers, we have compiled and synthesized literature on nitrogen removal by eastern oyster aquaculture, to support the development of a tool for use by both shellfish farmers and managers within the aquaculture permit review process. We have assessed the geographic scale at which nitrogen removal calculations can be based, and explored the potential for common cultivation practices and ploidy to influence the calculation of nitrogen removal at the farm scale. Based on this assessment, we have developed a publicly available, simple online tool that accurately predicts harvest-based nitrogen removal from an eastern oyster farm located within the geographic range of North Carolina to Maine, USA. The tool outputs were specifically designed for ease of integration into the US Army Corps of Engineers permit application review process. The approach and tool developed here are highly transferable to other shellfish species and regions globally where sufficient data exists to support a robust analysis. We have taken an adaptive management approach to tool development, basing our tool on current best available scientific information, with the explicit intention to maintain and update this tool when new information and data become available in the future.

## Materials and methods

### Data compilation

The Aquaculture Nutrient Removal Calculator (ANRC) is based on a publicly-available dataset of eastern oyster (*Crassostrea virginica*) morphometrics and nutrient concentration of tissue and shell measured on individual animals across the Northeast region of the United States, defined here as spanning the US states of North Carolina to Maine [31]. The ANRC was constrained to this geographic range for several reasons: 1) *C. virginica* is the sole oyster species cultivated across the entire region; 2) similar cultivation practices are employed across this region, including bottom cultivation without gear, off-bottom and/or floating aquaculture gear; 3) environmental conditions on farms are broadly similar, typically involving temperate estuaries and nearshore coastal locations in state waters.

The data compilation effort focused on assembling a dataset with geographic representation across the full region, and inclusion of all major cultivation practices (Fig 1; Table 1). Identification of data from each state was conducted to assemble a dataset most closely reflecting on-farm conditions as possible: Highest priority was given to identifying data collected from oysters that were grown on farms. If no data from farms were found, the second priority was

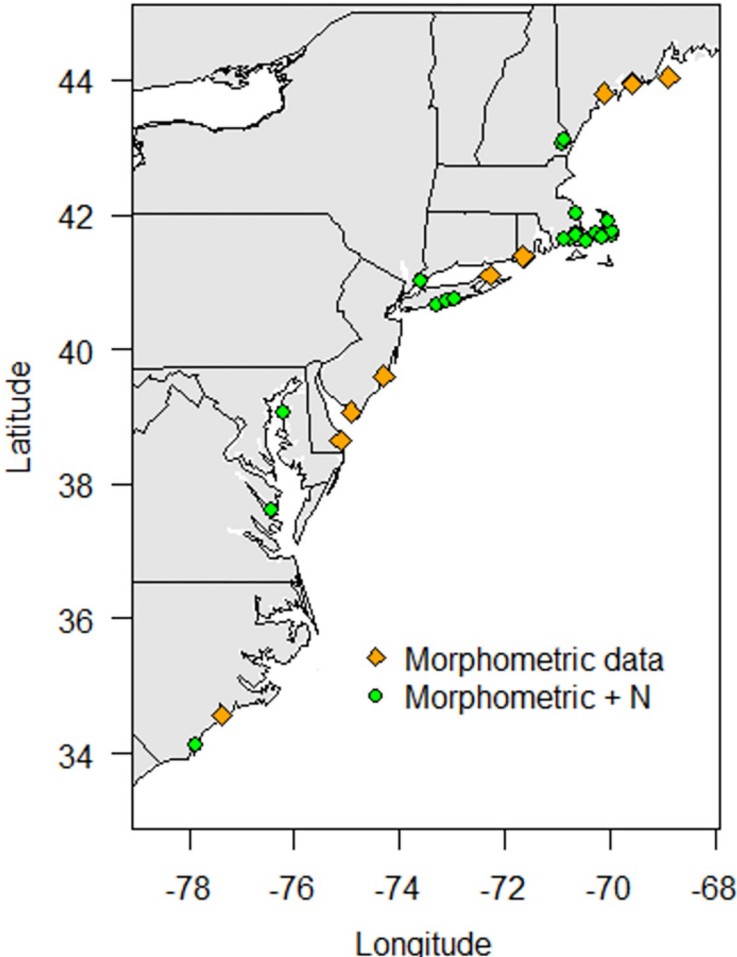

**Fig 1. Sample locations of eastern oyster (*C. virginica*) used in the present study from aquaculture farm sites along the eastern seaboard of the United States.** Orange diamonds show locations with morphometric data only, green circles show locations with both morphometric and nitrogen (N) concentration data.

identifying data collected from oysters that were grown by scientists employing common cultivation practices (e.g. hatchery-produced oysters, use of aquaculture gear, etc.). Most of the data fit one of these two categories, but for one state within the region (Rhode Island), data collected on wild oysters from waterbodies that had active oyster farms were used instead.

Data were identified using Web of Science searches related to oyster aquaculture in the Northeast United States, and by directly contacting scientists conducting oyster research within each state across the region. A comprehensive and exhaustive literature review of all published eastern oyster nutrient and morphometrics data was beyond the scope of this project, and most of the available peer-reviewed literature for eastern oysters has been collected on wild populations, which was not the focus of this tool. Most of the data in this dataset were previously published solely as summary statistics, and the raw data for individual animals was contributed by project leads to the new publicly available dataset (Fig 1; Table 1; [31]).

## Calculation of nitrogen removed at harvest

The approach employed here is an adaptation of methodology used by the Chesapeake Bay Program (CBP), within their nutrient management program, to calculate nitrogen reduction

**Table 1. The number of oyster samples by state, ploidy, and cultivation practice used for determining morphometric relationships (length:weight only) and nitrogen content (length:weight + nitrogen) for both tissue and shell material.** The asterisk indicates where the number of samples used in this report is an average of subsamples and relationships to morphometrics are indirect.

| Factor | | Tissue length:weight only | Tissue paired length:weight + nitrogen | Shell length:weight only | Shell paired length:weight + nitrogen |
|---|---|---|---|---|---|
| *State* | Maine | 1657 | 0 | 1657 | 0 |
| | New Hampshire | 0 | 199 | 199 | 0 |
| | Massachusetts | 75 | 152 | 75 | 152 |
| | Rhode Island | 108 | 0 | 108 | 0 |
| | Connecticut | 225 | 334 | 345 | 214 |
| | New York | 1480 | 40 | 1520 | 0 |
| | New Jersey | 144 | 0 | 144 | 0 |
| | Delaware | 225 | 334 | 345 | 214 |
| | Maryland | 14 | 294 | 240 | 68 |
| | Virginia | 4 | 309 | 253 | 60 |
| | North Carolina | 1965 | 11* | 1965 | 0 |
| *Ploidy* | Diploid | 4496 | 1107 | 5688 | 415 |
| | Triploid | 657 | 312 | 890 | 79 |
| *Cultivation Practice* | Bottom gear | 797 | 874 | 1471 | 200 |
| | Floating gear | 4002 | 111 | 4113 | 0 |
| | No gear | 854 | 434 | 994 | 294 |

by farmed eastern oysters in Chesapeake Bay, USA [27, 32]. Briefly, the CBP convened an expert panel to conduct a literature review and synthesis to determine the mean nitrogen concentration of eastern oyster tissue (% of dry weight), as well as robust regressions for the relationship between eastern oyster shell height and oyster tissue dry weight. The panel compiled data on nitrogen concentration in oyster tissue across the US Northeast region (Virginia to New Hampshire). Data on oyster morphometrics were constrained to the Chesapeake Bay only. Both nitrogen concentration and oyster morphometrics datasets included a mix of wild, restored, and farmed oysters. The CBP only considered nitrogen reduction associated with oyster tissue, and did not include nitrogen from oyster shells in their calculation. Farm-scale nitrogen removal at harvest was calculated by multiplying the mean nitrogen concentration of eastern oyster tissue by the tissue dry weight at the mean harvest size to calculate the weight of nitrogen in an individual oyster, then multiplying this value by the number of oysters harvested.

We employed the same methodology to our farm-based, region-wide data set of oyster tissue, and we extended the analysis here to include oyster shell, for calculation of whole-animal weight of nitrogen (g). Data were compiled as described above for shell nitrogen concentration (%), oyster tissue and shell dry weight (g), and oyster shell height (mm). Oyster shell height was defined as the longest distance (parallel to the long axis) between the hinge and the lip of the oyster; it is worth noting that some studies included in the regional data set referred to this measurement as oyster shell length. Information on oyster ploidy (diploid/triploid), harvest/sampling location, and cultivation practice (grouped into three categories: no gear, bottom gear, floating gear) were also recorded for each individual oyster in the dataset.

Hypothesis testing was conducted using the statistical software program R version 4.3.1 (www.r-project.org). Nutrient concentration of eastern oyster tissue and shell was compared across states, ploidy, and cultivation practice. Due to data gaps in cross-factor datasets, 1-way hypothesis testing was chosen over a two-way analysis for main effects and interactions. A

robust analog of ANOVA was employed that used a percentile bootstrap method with 20% trimmed means to compare independent groups [33]. The mean nitrogen concentration of oyster tissue and shell was calculated for the full regional dataset, and this mean value was applied in calculations of nitrogen removal across all locations, cultivation practices, and ploidy. The extreme curvature in the relationship between oyster dry weight and shell height was addressed in hypothesis tests by natural log transformation of shell height and oyster dry weight (tissue and shell) to approximate linearity, an approach previously employed in the shellfish aquaculture literature (e.g. [34, 35]). Robust regression analysis was conducted on the transformed data using the Theil-Sen regression estimator and a percentile bootstrap method [36]. While a transformation was applied in order to facilitate hypothesis testing, nonlinear methodology could be used within the tool itself to describe the morphometrics relationships on the original untransformed data. Nonlinear quantile regressions were generated using the R statistical package quantreg [37, 38], based on the previous literature indicating that the relationship between oyster dry weight and shell height is described by a power function [27]. The 50th quantile was used as an estimate of the median of the dataset, as 50% of the tissue dry weight values lie above each value of shell height using this approach. The use of nonlinear quantile regression also facilitates direct comparison to the previous Chesapeake Bay oyster nutrient best management practice [27].

## Tool development

The ANRC is a tool designed for growers and resource managers to inform shellfish aquaculture permitting. Resource managers have expressed interest in easy-to-use tools that produce location and operation-appropriate values for beneficial services, and these values need to be produced in a format that aligns with the permitting process. We have synthesized available literature for eastern oyster farms across the US Northeast region (North Carolina to Maine), and applied methodology used by the Chesapeake Bay Program to calculate nutrient removal at harvest. Variability in oyster tissue and shell nutrient concentration was low, and an assessment of farm location, ploidy, and cultivation practice (with vs. without gear) suggested that a single average value could reasonably be applied across all farms.

The nutrient removal calculations are based on relationships of oyster dry weight-to-length and the average nitrogen (N) concentrations in shell and tissue material. First, we estimate the weight of the oysters based on the typical size of oysters harvested on a farm. The weight estimates are based on non-linear quantile regressions of oyster shell height and dry-weight for both tissue and shell material. Next, the N portion of total oyster weight is calculated using the average N concentration value for both shell and tissue. Adding the tissue and shell nitrogen yields the total weight of N per oyster. This result is scaled to the total number of oysters harvested, as input by the user.

The ANRC can be used for new permit applications based on estimated production value or to provide information on existing farms from actual harvest numbers. The tool was developed as an R Shiny application and is hosted on the NOAA Posit Connect server, hosted here: https://connect.fisheries.noaa.gov/ANRC/

The application (version 1.0) is divided into three tabs, the 'Calculator'tab is the landing page, where the user is able to input their information and generate a report, a 'Reverse Calculator'tab, where a user can calculate the number of oysters required to mitigate a given nitrogen load, and an 'About'tab that explains the project background and user inputs, and contains details on data contributors, sample locations, and references.

The Calculator inputs are separated into three categories: Farm Practices, Farm Location, and Harvest Details. Farm practices include the name of the farm, oyster ploidy, and culture

method primarily used for growing oysters. These values currently do not affect the calculation of nitrogen removal, and are simply incorporated into the report.

Farm Location input includes a text box for the harvest location, and a Leaflet-based map interface where the user can drop a location pin on the waterbody where the oysters were harvested. The map interface records the latitude and longitude of the dropped marker.

Harvest Details include the average oyster size at harvest, the total number of oysters harvested, and the period of harvest. Our analysis of the region-wide values of nutrient concentration and morphometric relationships for oyster shell and tissue suggest that just the number and size of the oysters harvested are sufficient to provide a robust region-wide estimate of nutrient removal associated with oyster harvest. The period of harvest (1 day to 5 years) is included for use in generating the report, but does not affect the calculation of nitrogen removed.

## Results and discussion

### Oyster tissue and shell nitrogen concentration

The mean nitrogen concentration of oyster tissue was 7.70% across the full regional dataset (Table 2; n = 1,339, sd = 1.34, range 2.59–14.08). The nitrogen concentration of oyster shell was 0.19% across the full regional dataset (Table 2; n = 494, sd = 0.10, range 0.04–0.86). These observations were similar to the previously reported mean of oyster tissue nitrogen concentration by the CBP (mean 8.2%, range 5.64–9.27%, n = 17 [27], and by [23] mean 8.22%, range 7.0–11.8%, n = 14). The overall range of observed values was slightly higher here than in these previous studies, but this was not surprising given the two orders of magnitude increase in sample size over previous work. Similarly, the mean nitrogen concentration of oyster shell reported here was very close to that reported by [23], (mean 0.23%, range 0.13–0.32%, n = 13), with a much larger sample size and overall larger range in observed values. While the CBP does not include oyster shell in their calculation of oyster aquaculture nitrogen reduction, they have included oyster shell in their calculation of nitrogen reduction associated with restored oyster reefs, and the data compiled here show the same trend of very similar mean, slightly greater range, much larger sample size ([39] mean 0.2%, range 0.08–0.32%, n = 15). The similarities in overall mean, with a much larger and geographically extensive dataset, strengthens previous evidence indicating low overall variation in nitrogen concentration of eastern oyster tissue and shell.

**Table 2. Nitrogen concentration of eastern oyster tissue and shell by location, ploidy, and cultivation practice.** Data are reported as mean (standard deviation).

| Factor | | Tissue N | Shell N |
|---|---|---|---|
| Full Regional Dataset | | 7.70 (1.34) | 0.19 (0.10) |
| *State* | New Hampshire | 7.54 (2.05) | ND |
| | Massachusetts | 8.01 (1.26) | 0.24 (0.14) |
| | Connecticut | 7.51 (1.02) | 0.14 (0.05) |
| | New York | 9.24 (0.51) | ND |
| | Maryland | 7.25 (0.98) | 0.17 (0.05) |
| | Virginia | 8.05 (1.23) | 0.24 (0.07) |
| | North Carolina | 8.76 (0.79) | ND |
| *Ploidy* | Diploid | 7.76 (1.41) | 0.18 (0.10) |
| | Triploid | 7.52 (1.10) | 0.22 (0.09) |
| *CultivationPractice* | Bottom gear | 7.66 (1.43) | 0.21 (0.09) |
| | Floating gear | 9.18 (0.61) | ND |
| | No gear | 7.62 (1.09) | 0.18 (0.10) |

**Table 3. Results of hypothesis testing for the effect of location, ploidy, and cultivation practice on tissue and shell nitrogen concentration within the regional data set.**

| Comparison | Tissue N | Shell N |
|---|---|---|
| Location | p < 0.001 | p < 0.001 |
| | Effect size 0.57 | Effect size 0.34 |
| Ploidy | p = 0.001 | p < 0.001 |
| | Effect size 0.29 | Effect size 0.04 |
| Cultivation Practice | p < 0.001 | p < 0.001 |
| | Effect size 0.23 | Effect size 0.04 |

Diploid and triploid oysters exhibited small but statistically significant differences in tissue and shell nitrogen concentration (Table 3; both p ≤ 0.001). The effect sizes were small relative to the magnitude of nitrogen concentration for both tissue and shell, as well as the overall range of nitrogen concentration observed here and in the literature (Figs 2C and 3C; [23, 27]; as summarized above). Mean tissue nitrogen concentration was 7.76% for diploid oysters, and

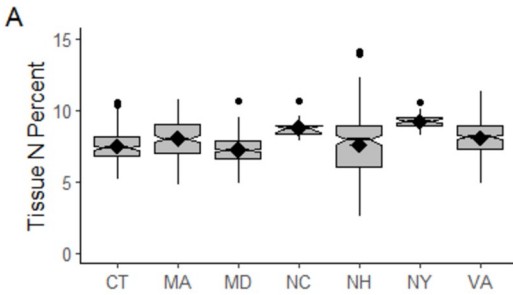

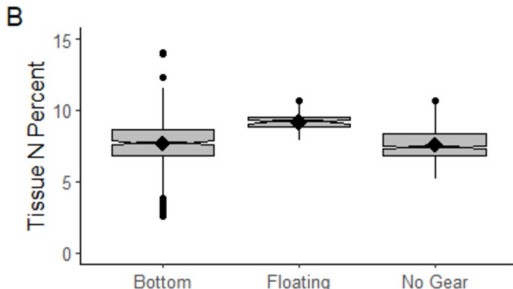

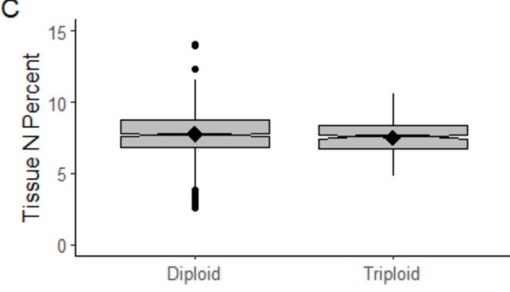

**Fig 2.** Summary of eastern oyster tissue percent nitrogen (N) composition by (A) state, (B) cultivation practice, (C) ploidy. Median values are illustrated by the notch and horizontal black lines in each boxplot, and mean values are shown as black diamonds. Circles indicate outlier values that fall outside 1.5x the interquartile range.

7.52% for triploids. Mean shell nitrogen concentration was 0.18% for diploids and 0.22% for triploids. In light of the small effect sizes observed between diploid and triploid oysters, we opted to use a single mean nitrogen concentration in the ANRC, regardless of the ploidy of harvested oysters.

Small but statistically significant differences in oyster tissue and shell nitrogen concentration were also observed across cultivation practices ($p < 0.001$) (Table 3), and effect sizes were also small relative to the magnitude of nitrogen concentration and the overall range of nitrogen concentration observed here and in the literature (Figs 2C and 3C; [23, 27]). Mean tissue nitrogen concentration was 7.66% for oysters cultivated in bottom gear, 9.18% for oysters cultivated in floating gear, and 7.62% for oysters cultivated on the seafloor with no gear. Mean shell nitrogen concentration was 0.21% for oyster cultivated in bottom gear, and 0.18% for oysters cultivated on the seafloor without gear. We were unable to find data on shell nitrogen concentration for oysters grown in floating gear in the published literature. The absence of shell nitrogen concentration data from floating gear does represent a data gap, and the number of data points available for oyster tissue nitrogen concentration was relatively low compared to bottom gear and no gear (floating gear n = 47, bottom gear n = 874, no gear n = 418).

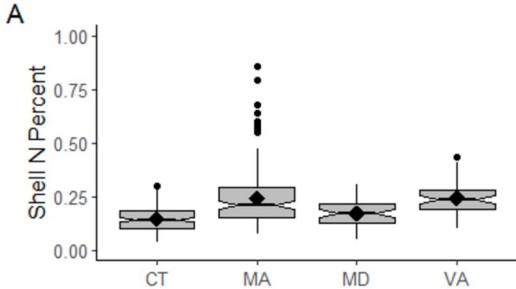

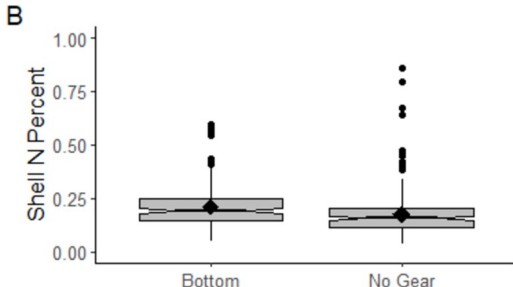

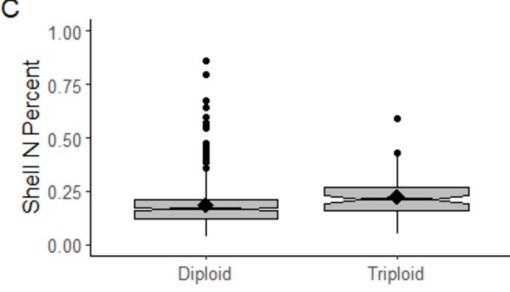

**Fig 3.** Summary of eastern oyster shell percent composition of nitrogen (N) summarized by (A) state, (B) cultivation practice, (C) ploidy. Median values are illustrated by the notch and horizontal black lines in each boxplot, and mean values are shown as black diamonds. Circles indicate outlier values that fall outside 1.5x the interquartile range.

However, given the overall low variation in shell nitrogen concentration across all other factors considered, we believe available information indicates that the regional tool can be used to accurately predict the nitrogen removal by farmed eastern oysters grown in floating gear. Additionally, the ANRC can be easily updated should new data become available in the future that indicates oysters grown in floating gear contain substantially different tissue or shell nitrogen concentrations. Adaptive management based on an evolving scientific literature is a common practice in nutrient management (e.g., [40, 41]), and can easily be accommodated by the ANRC.

Small but statistically significant differences in oyster tissue and shell nitrogen concentration were observed based on a global test of location (both p < 0.001) (Table 3), with small effect sizes relative to the magnitude of nitrogen concentration and the overall range of nitrogen concentration observed here and in the literature (Figs 2C and 3C; [23, 27]). We opted not to perform pairwise comparisons given the low statistical power associated with controlling for familywise error across 21 pairs. The range in mean tissue nitrogen concentration across states was 7.25–9.24% and the range in mean shell nitrogen concentration across states was 0.14–0.24. Shell nitrogen concentration is more rarely reported in the literature, and data were only available for 4 states (VA, MD, CT, MA). Given that these states span both the mid-Atlantic and New England sub-regions of the US Northeast, and given the low observed variation, we believe a region-wide mean shell nitrogen concentration is supported by the available data.

## Oyster morphometrics

The 50th quantile nonlinear power function provided a good fit to both the tissue and shell datasets (Fig 4). The regression equation for tissue dry weight vs. shell height was $y = 1.42e^{-5}x^{2.607}$ for the full regional dataset. The regression equation for shell dry weight vs. shell height was $y = 0.00039x^{2.58}$ for the full regional data set.

Diploid and triploid oysters exhibited small but statistically significant differences (all p < 0.05) in the relationship between transformed shell height and tissue/shell dry weight (Table 4; Fig 5). The effect sizes were small: 0.2 (tissue slope), 0.3 (tissue intercept), 0.13 (shell slope), and 0.79 (shell intercept) and the differences in tissue and shell dry weight were particularly small at the typical size range that eastern oysters are harvested in the Northeastern US (2.5–3.5 inches, 63.5–88.9 mm; Fig 5 vertical lines). These results are inconsistent with the current nutrient management practice by the CBP that assigns a higher tissue dry weight to triploid oysters across the range of oyster shell heights [27]. The analysis conducted by the CBP in support of their nutrient management program did not contain sufficient data to allow direct comparison of diploids and triploids grown under identical conditions, and all of the triploid data in the CBP dataset came from a single study [27, 32]. A more recent study that directly compared morphometrics of diploid and triploid eastern oysters grown in bottom cages at two farms in Maryland and Virginia, USA found no significant difference in the relationship between shell height vs. tissue dry weight between diploids and triploids at either farm individually or when the farm data was combined [35]. The data from [35] are included in the ANRC dataset, as well as data from two farms in Maine, USA, that grew diploids and triploids in floating gear, and had very small effect sizes in the relationship between shell height vs. tissue dry weight by ploidy. Due to the small ploidy-based difference in oyster morphometrics when oysters were grown in either bottom or floating gear, as well as the absence of data supporting a difference in diploids and triploid when grown under identical conditions, we have opted to use a single regression for both diploid and triploid oysters in the ANRC tool.

Small differences in oyster morphometrics were also observed based on cultivation practices, with a combination of significant and not significant statistical results (Table 4; Fig 6).

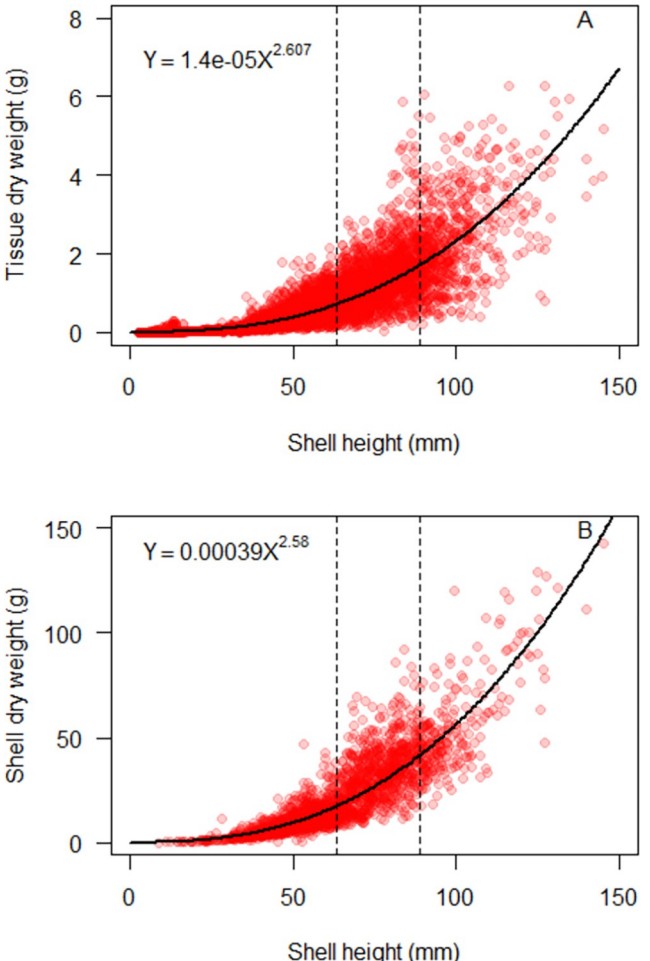

**Fig 4.** The relationship between eastern oyster dry weight and shell height for (A) tissue and (B) shell. (Solid black line) 50th quantile regression of a nonlinear power function for the ANRC regional data compiled in this study. Dashed vertical lines represent typical market size for harvested eastern oysters in the Northeastern US, ranging from 2.5–3.5 in (63.5–88.9 mm).

There were no significant differences in oyster morphometrics between the bottom gear and no gear cultivation practices, for either tissue or shell datasets (all p > 0.05). Small but significant differences (all p < 0.05) were observed between floating gear vs. no gear cultivation practices, with effect sizes 0.80 (tissue slope), 3.6 (tissue intercept), 0.21 (shell slope), and 1.1 (shell intercept). Small but significant differences (all p < 0.05) were also observed between floating gear vs. bottom gear cultivation practices, with effect sizes 0.69 (tissue slope), 2.9 (tissue intercept), 0.18 (shell slope), and 0.84 (shell intercept). The absence of a significant result between bottom gear and no gear cultivation practices clearly supports the use of a single regression to represent these two cultivation practices. While there were significant differences between floating gear and the other two cultivation practices, the size of this effect was minimal, particularly at the typical size range that eastern oysters are harvested in the Northeastern US (2.5–3.5 inches, 63.5–88.9 mm; Fig 6 vertical lines). Differences were more pronounced at oyster shell heights exceeding 125 mm (~5 inches), but as eastern oysters are rarely harvested at this size, we did not feel this justified the development of a more complex tool. As observed with the nitrogen concentration data, the ANRC can be easily updated should new data become

**Table 4. Results of hypothesis testing for the effect of ploidy, cultivation practice, and regions on the relationship between oyster shell height vs. oyster tissue and shell dry weight.**

| Comparison | Tissue slope | Tissue y-intercept | Shell slope | Shell y-intercept |
|---|---|---|---|---|
| Diploid (D) vs. triploid (T) | $p < 0.001$ | $p < 0.001$ | $p = 0.04$ | $p < 0.001$ |
|  | D = 2.74 | D = -11.7 | D = 2.75 | D = -8.55 |
|  | T = 2.58 | T = -11.4 | T = 2.88 | T = -9.35 |
| No gear (NG) vs. bottom gear (BG) | $p = 0.29$ | $p = 0.09$ | $p = 0.66$ | $p = 0.47$ |
|  | NG = 2.03 | NG = -8.44 | NG = 2.60 | NG = -7.84 |
|  | BG = 2.14 | BG = -9.26 | BG = 2.63 | BG = -8.09 |
| No gear (NG) vs. floating gear (FG) | $p < 0.001$ | $p < 0.001$ | $p < 0.001$ | $p < 0.001$ |
|  | NG = 2.03 | NG = -8.44 | NG = 2.60 | NG = -7.84 |
|  | FG = 2.83 | FG = -12.1 | FG = 2.81 | FG = -8.93 |
| Floating gear (FG) vs. bottom gear (BG) | $p < 0.001$ | $p < 0.001$ | $p <0.001$ | $p < 0.001$ |
|  | FG = 2.83 | FG = -12.1 | FG = 2.81 | FG = -8.93 |
|  | BG = 2.14 | BG = -9.26 | BG = 2.63 | BG = -8.09 |
| New England (NE) vs. Mid-Atlantic (M) | $p < 0.001$ | $p < 0.001$ | $p < 0.001$ | $p < 0.001$ |
|  | NE = 2.89 | NE = -12.3 | NE = 2.94 | NE = -9.3 |
|  | M = 2.54 | M = -10.9 | M = 2.66 | M = - 8.3 |

available in the future that indicates oysters grown in floating gear exhibit a substantially different relationship between tissue/shell dry weight vs. shell height.

While the full regional dataset had good spatial coverage, containing morphometrics data from all states, within-state data was more limited (Table 1). Of the 11 states with oyster morphometrics data available, only 4 had sufficiently large datasets to generate a statistically significant quantile regression for the relationship between oyster tissue dry weight and oyster shell height (North Carolina, New York, Connecticut, Maine), and only 2 had sufficient data to generate a statistically significant quantile regression for the relationship between oyster shell dry weight and oyster shell height (North Carolina, Maine). Within these states, geographic coverage was limited, with 3 farms sampled across 2 estuaries in North Carolina, 4 locations sampled across 2 estuaries in New York, 1 location sampled in Connecticut, and 4 locations sampled across 3 estuaries in Maine. None of the state-specific farmed oyster datasets contained the within-estuary spatial resolution achieved by the CBP in their mixed wild/restored/farmed oyster dataset, which had samples from 22 locations across the upper, mid, and lower Bay, and included the full range of salinity conditions across which oysters are grown in this estuary [27]. It is unclear whether oysters sampled from a small number of locations within a single coastal waterbody adequately capture the potential variability in this relationship within the waterbody/state as a whole.

While state-specific comparisons in oyster morphometrics were not feasible, we did assess the potential for a general north-south trend by dividing the dataset into two geographic sub-regions (Fig 7). Oyster morphometrics data from the states of Maine, New Hampshire, Massachusetts, Rhode Island, and Connecticut were pooled into a "New England" dataset, and compared to data from the "Mid-Atlantic", which included the states of New York, New Jersey, Delaware, Maryland, Virginia, and North Carolina. The comparison was statistically significant, but the effect sizes were small: 0.35 (tissue slope), 1.4 (tissue intercept), 0.28 (shell slope), and 1.0 (shell intercept) and the differences in tissue and shell dry weight were particularly small at the typical size range that eastern oysters are harvested in the Northeastern US (2.5–3.5 inches, 63.5–88.9 mm; Fig 7 vertical lines).

At the same time, the range in observed values for oyster tissue dry weight across the US Northeast region was similar to the previously reported range for oyster tissue dry weight

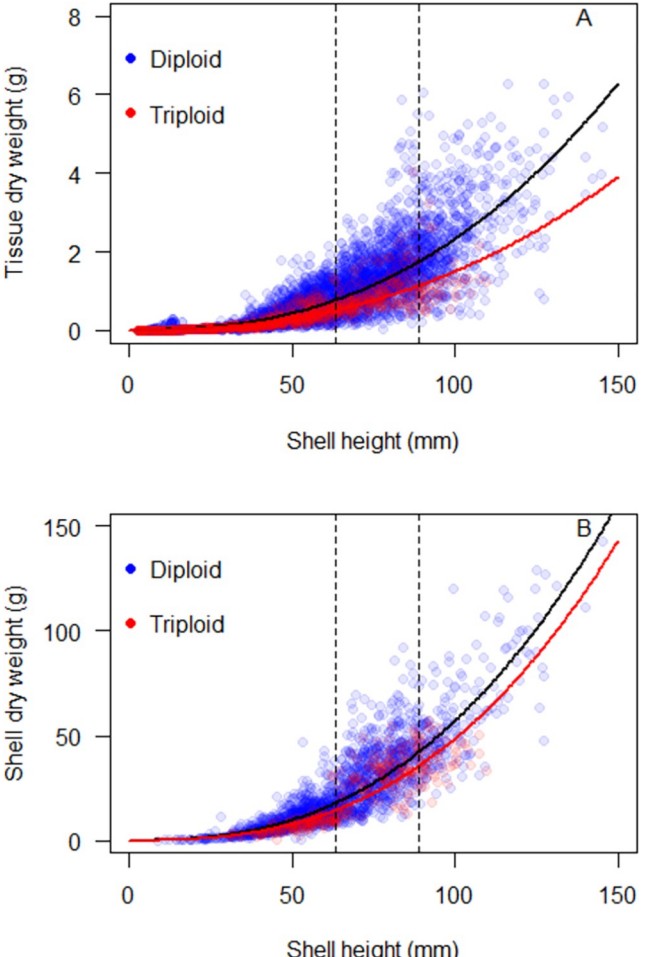

**Fig 5.** Eastern oyster dry weight (g) vs. shell height (mm) for tissue (A) and shell (B) for diploid oysters (blue circles, black regression line) and triploid oysters (red circles, red regression line). Regression lines are the 50th quantile fit of a nonlinear power function. Dashed vertical lines represent typical market size for harvested eastern oysters in the Northeastern US, ranging from 2.5–3.5 in (63.5–88.9 mm).

within Chesapeake Bay alone, across all oyster shell heights measured (Fig 8A). This result indicates that the variability in oyster tissue dry weight across the region as a whole was similar to the variability in oyster tissue dry weight within a single, well-sampled estuary. The similarity in the ranges of these datasets suggests that the sample size of each may be sufficient to reflect the physiological limits of the relationship between tissue dry weight vs. shell height for eastern oysters given environmental conditions in northern temperate estuaries.

The range of tissue dry weight observed within the regional, farm-based dataset was similar to that observed for the CBP dataset's mixture of wild, farmed, and restored oysters. This similarity was a little surprising, given the efforts by many farmers to produce oysters with higher meat weight [42]. While the overall range is visually comparable, unfortunately, the central tendency in the data could not be easily compared, as the Chesapeake Bay Program separated their reporting of 50th quantiles by ploidy, and approximately 20% of the data used in the Chesapeake Bay analysis has not been made publicly available [32]. It would be worthwhile to compare the average tissue dry weight of the regional, farmed dataset to the average tissue dry weight of the CBP data if all of the data become available in the future.

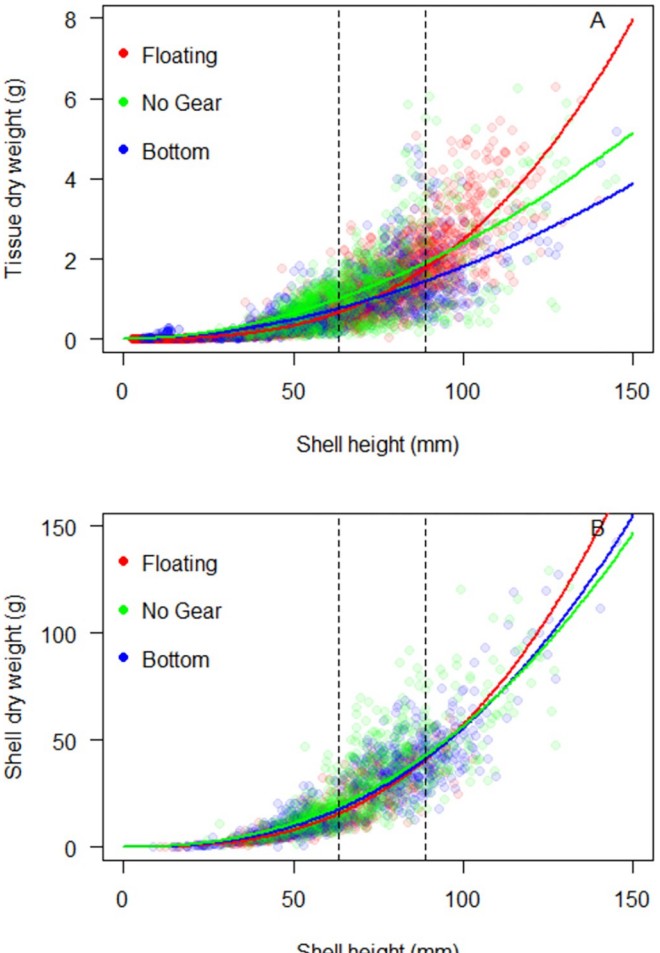

**Fig 6.** Eastern oyster dry weight vs. shell height for tissue (A) and shell (B) for eastern oysters cultivated in floating gear (red), no gear (green), and bottom gear (blue). Regression lines represent the 50th quantile fit of a nonlinear power function. Dashed vertical lines represent typical market size for harvested eastern oysters in the Northeastern US, ranging from 2.5–3.5 in (63.5–88.9 mm).

The range in observed values for oyster shell dry weight across the US Northeast region was less than the previously reported range for oyster tissue dry weight within the Chesapeake Bay, across all oyster shell heights measured (Fig 8B). In general, it appears that the oysters within the regional dataset had lighter shells at a given size than oysters from the CBP, although again, direct comparison of 50th quantile regressions was not possible due to reporting differences and incomplete data. Reduced shell weights of farmed oysters have been reported previously in the literature, particularly for oysters grown in gear [42], which would be consistent with the trend observed here. While the range of tissue dry weights were similar between our farm-centric, regional data and the mixed wild/restored/farmed oyster data from Chesapeake Bay, the difference in shell weights between the two datasets highlights the value of the farm-specific approach taken here. It appears that the use of shell data from a dataset containing wild/restored oysters to calculate nitrogen removal on farms could result in an overestimate of the nitrogen removal service provided.

After consideration of the similarity in range of reported oyster tissue/shell dry weights between the Chesapeake-only and the full regional datasets, the small effect size of the

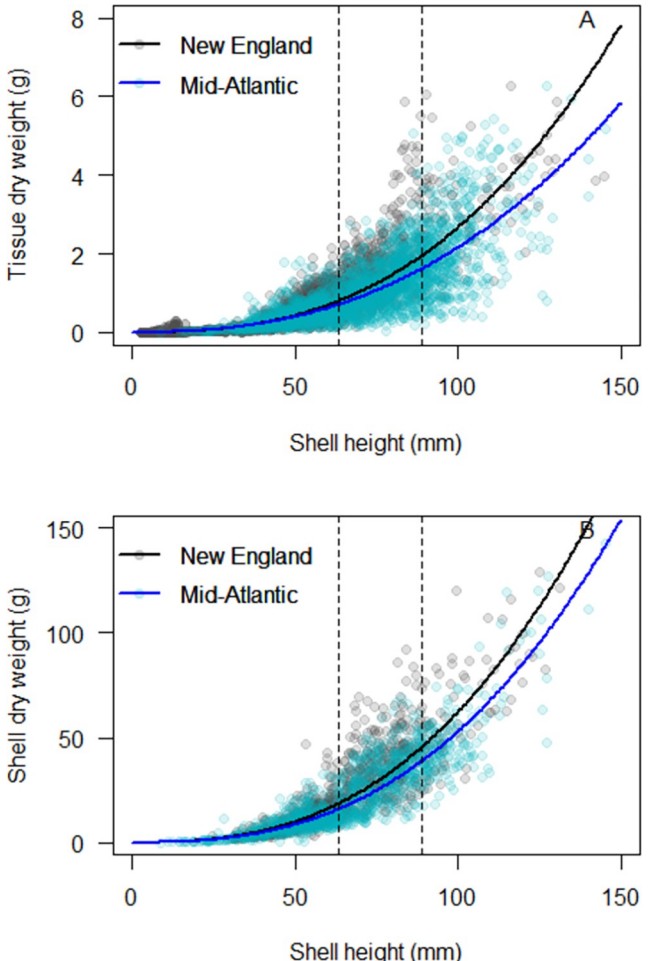

**Fig 7.** Eastern oyster dry weight vs. shell height for tissue (A) and shell (B) for the New England (gray) and Mid-Atlantic (blue) sub-regions of the US Northeast. Regression lines represent the 50th quantile fit of a nonlinear power function. Dashed vertical lines represent typical market size for harvested eastern oysters in the Northeastern US, ranging from 2.5–3.5 in (63.5–88.9 mm).

comparison between New England vs. Mid-Atlantic geographic sub-regions, and the geographic limitations of the within-state data, we opted for a single tissue regression and a single shell regression across the full region. We believe this choice is a conservative approach that is well supported by available data, and this approach can be easily revisited in the future as new data is collected.

## Calculation of nitrogen removal

As described above, low variation was observed in eastern oyster nitrogen concentration and morphometrics related to ploidy, cultivation practice, and location within the regional data set. Our analysis indicated that the magnitude of these differences was not sufficient to justify a more complex tool. Furthermore, discussions with industry stakeholders at a regional aquaculture conference (Northeast Aquaculture Conference and Expo 2024) and at a national aquaculture conference (Aquaculture America 2024) indicated industry preference for the simplest possible tool, in order to facilitate the broadest implementation for farms across the region. A single regional value for nitrogen concentration of oyster tissue and shell, a single regional

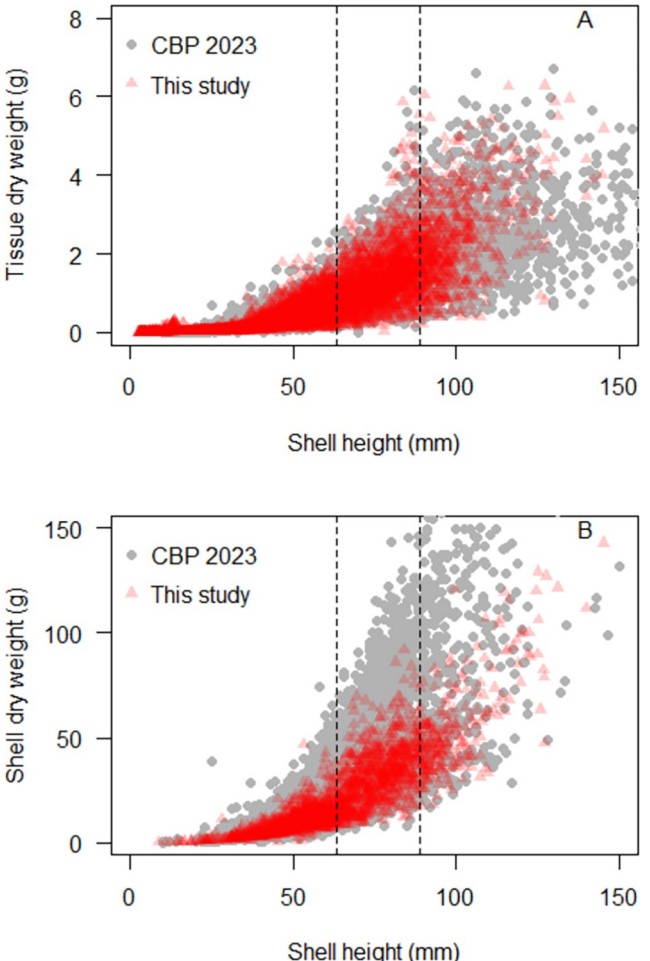

**Fig 8.** The relationship between eastern oyster dry weight and shell height for (A) tissue and (B) shell. (Red triangles) ANRC regional data spanning the US east coast from Maine to North Carolina. (Gray circles) previously reported oyster morphometrics data for wild, farmed, and restored oysters in Chesapeake Bay.

regression describing the relationship between oyster tissue dry weight vs. shell height, and a single regional regression describing the relationship between oyster shell dry weight vs. shell height have been employed here to calculate mean nitrogen removal by farms across the US Northeast Region (North Carolina to Maine).

In the ANRC online tool, the weight of nitrogen for an individual oyster was calculated by multiplying the 50th quantile of tissue dry weight (given the mean oyster shell height at harvest; an input variable) by the regional mean tissue nitrogen concentration to obtain the tissue nitrogen (g). This process was repeated with shell dry weight and mean shell nitrogen concentration to calculate the weight of shell nitrogen (g). The two values were added together to obtain whole animal nitrogen (g). Whole animal nitrogen was then multiplied by number of individuals harvested, as input by the user, to obtain farm-scale nitrogen removal, which can be reported in either pounds or kilograms.

Nitrogen removal outputs for the harvest of 1 million oysters were calculated using the ANRC online tool, for three common sizes of harvested eastern oysters (2.5, 3, 3.5 in; Table 5). Nitrogen removal was 194 pounds for 2.5 in oysters (interquartile range 143–274), 311 pounds for 3 in oysters (IQR 234–274), and 465 pounds for 3.5 in oysters (IQR 354–611). The use of a

**Table 5. Potential nitrogen removal in pounds (lbs) associated with the harvest of one million oysters at four different sizes using regressions based on either 1) cultivation practice (light gray shading), 2) ploidy (dark gray shading), or 3) an overall regression using all data (no shading).** The interquartile range is shown for the overall regression based on all farm data across the region.

| | | Shell Height: mm (inches) | | |
|---|---|---|---|---|
| **Factor** | **Class** | **64 (2.5)** | **77 (3)** | **89 (3.5)** |
| *Cultivation Practice* | *Bottom* | 209 | 307 | 427 |
| *Cultivation Practice* | *No Gear* | 259 | 375 | 515 |
| *Cultivation Practice* | *Floating* | 184 | 310 | 482 |
| *Ploidy* | *Diploid* | 210 | 330 | 483 |
| *Ploidy* | *Triploid* | 150 | 236 | 347 |
| ***Overall*** | | 194 | 311 | 465 |
| *Inter-quartile range:* | | *143—274* | *234—423* | *354—611* |

single regression equation for oyster morphometrics on nitrogen removal was evaluated by calculating nitrogen removal using different regressions for cultivation practice (bottom vs. no gear vs. floating gear) and for ploidy (diploid vs. triploid; Table 5). When cultivation practice was varied for 3 inch oysters, the differences in nitrogen removal were very small (<4%) among the overall regression (311 pounds), floating gear (310 pounds), and bottom gear (307 pounds). Use of the overall regression resulted in a smaller calculation of nitrogen removal than would have been predicted by the regression for oysters grown without gear (311 vs. 375 pounds), but all of these values were well within the interquartile range for the full regional dataset (234–423 pounds). When ploidy was varied for 3 inch oysters, the differences in nitrogen removal were larger: 311 pounds for the overall regression, 330 pounds for diploids, and 236 pounds for triploids, however, the ploidy-based differences were again within the interquartile range for the full regional regression. We do not believe that these differences are large enough to justify the development of a more complex tool at this time.

## Integration with shellfish aquaculture permit review

Due to the results of the analysis supporting the use of a single nitrogen concentration across all locations and farming practices, as well as a single regression to describe the relationship between oyster shell height and tissue/shell dry weight, the development of a simple tool to calculate nitrogen removal at the farm scale is complementary to the ease of integration into the permit review process for shellfish aquaculture in the US Northeast. The comprehensive scale of the analysis provides support for defensible integration of consideration of nitrogen removal into permitting review decisions. To further enhance ease of integration into the permitting process, we have designed the ANRC tool to both generate nitrogen removal outputs that display online in numerical and graphical format, and also in a downloadable report that can be attached to permit application materials. The report is tailored for integration with the US Army Corps of Engineers public interest review process and similar state-level permitting processes, and provides a succinct summary of the ecological services associated with nutrient removal in eutrophic locations, project-specific values, and citations supporting the calculation of those values. In discussion with US Army Corps of Engineers project managers, this information can be specifically integrated into their public interest review related to water quality effects from a proposed project. Future iterations of the tool may include information related to the eutrophication status of project areas, further emphasizing the nutrient removal service of shellfish aquaculture in impaired waters.

Project-specific considerations in the report include farm location (general description as well as GIS coordinates), harvest period, number of oysters harvested, and size of harvested oysters. This allows growers to account for nitrogen removal between different-sized oysters at harvest. As nitrogen removal is tied to the size of oysters, and oyster size at harvest may vary within and across farms, based on either market preferences and/or state regulations on minimum harvest size, the tool allows for growers to account for these differences and run individual calculations based on varying oyster harvest sizes across and/or within different harvest periods. The adjustable timeframe in the tool also provides information for managers that may want to equate nitrogen removal to a particular timeframe for an authorized permit or lease.

## Comparison to other shellfish ecosystem services calculators

The ANRC joins a growing number of shellfish ecosystem services calculators available online, while providing a unique combination of functionality and geographic range. The Nature Conservancy maintains an online Oyster Calculator (https://oceanwealth.org/tools/oyster-calculator/) for restored oyster reefs, based on peer-reviewed syntheses of the scientific literature. The Oyster Calculator provides the volume of water filtered for locations around the United States using methods described in [43], and subsequently expanded with environmental data reported in [44]. Custom locations outside the existing database can also be used if data on bay volume, residence time, and temperature are provided by the user. The Oyster Calculator also outputs enhanced production of wild fish and invertebrates from habitat provided by restored oyster reefs for the Northern Gulf of Mexico, Floridian, Carolinian, and Virginian ecoregions, using methods described in [44]. This calculator is focused on restored oyster reefs, and does not currently calculate oyster nitrogen removal.

Rutgers University maintains an online Oyster Eco-Serve Calculator (https://itsappserver.sebs.rutgers.edu/Oysterecoservecalc/) for eastern oyster farms from Massachusetts to Virginia, USA. The Oyster Eco-Serve Calculator provides the volume of water filtered, the total mass of particles removed through oyster filtration, and the total mass of particles added to the ecosystem through oyster waste, on an annual, seasonal, and daily basis. The data and methods supporting the calculator are described in [7]. All of the data supporting this calculator were collected in the field on working oyster farms, but the calculator does not currently calculate oyster nitrogen removal at harvest.

The University of Florida maintains an online Florida Shellfish Farm Nitrogen Calculator (https://ufl864.outgrow.us/ufl864-2) for eastern oyster and hard clam (*Mercenaria mercenaria*) farms within the state of Florida, USA. The calculator outputs nitrogen removal from shellfish harvest, from enhancement of denitrification in sediments underneath farms, and a potential monetary value of the combined nitrogen removal service. All of the data for oyster harvest nitrogen were collected from farmed oysters in Florida, the denitrification enhancement was measured in the field on working oyster and clam farms, and the value of this service was calculated based on a willingness to pay survey of Florida wastewater treatment plants. This calculator is currently intended only for use by Florida oyster and clam farms.

All of these calculators provide different and valuable information on shellfish ecosystem services, based on robust scientific data collection. The ANRC adds to this existing body of work by providing a robust calculation of the typical expected harvest-based nitrogen removal for any eastern oyster farm from North Carolina to Maine, USA, and by generating outputs that are intended to directly integrate into state and federal shellfish aquaculture permitting processes that allow for consideration of environmental benefits in the permit application review. We compared our results to the Florida Shellfish Farm Nitrogen Calculator using inputs of 1,000,000 oysters harvested at a shell height of 3 inches. The Florida Calculator

predicted nitrogen removal of 265 pounds, and the ANRC tool predicted 311.6 pounds of nitrogen. We also compared our tissue-only nitrogen removal results to the Chesapeake Bay diploid regression. The Chesapeake Bay diploid tissue regression predicted nitrogen removal of 198 pounds, and the ANRC tool predicted 194.5 pounds.

The approach described here is highly transferable to other locations and other shellfish species for which sufficient data exists to predict nitrogen removal by shellfish farms with high confidence. Given the low observed variation in oyster tissue and shell nitrogen concentration across the geographic region, ploidy, and common cultivation practices in this study, the data needed to accurately quantify whole animal nitrogen concentration for other species and locations should be relatively modest. While data associated with oyster morphometrics were more variable, other shellfish species may exhibit lower variability in weight-to-length ratios than oysters, potentially reducing data needs. Furthermore, measurements of dry weight and length are much less expensive and less technologically challenging than measurements of nitrogen concentration, making these data easier to obtain.

## Conclusions

The ANRC is a valuable new tool for the shellfish aquaculture industry and for the shellfish aquaculture permit review process, and is directly responsive to management needs. The size and breadth of the dataset compiled in this study gives high confidence in the farm-scale harvest nitrogen removal calculations, and the report generated is designed to be easily integrated into the permit application for a new or expanding farm. The high transferability of the approach employed here increases the potential future application of this work to a variety of bivalve shellfish species farmed in eutrophic locations around the world.

## Supporting information

**S1 Fig. Oyster shell height to dry-weight interquartile range for tissue (A) and shell material (B) based on cultivation practice.** The median values are shown as bold lines. (TIF)

## Acknowledgments

The authors would like to thank all of the data sources who generously contributed their raw data to aid in the development of the ANRC tool. Datasets originated from Damian Brady and Tom Kiffney (University of Maine), Ray Grizzle and Krystin Ward (University of New Hampshire), Josh Reitsma (Cape Cod Cooperative Extension), Suzy Ayvazian (EPA Narragansett Lab), Skylar Bayer, Matt Poach, Shannon Meseck, and Julie Rose (NOAA Milford Lab), Jeff Levinton, Michael Doall, and Daria Sebastiano (Stony Brook University), Daphne Munroe and Janine Barr (Rutgers University), Julie Reichert-Nguyen (NOAA Chesapeake Bay Office), Suzanne Bricker (NOAA Oxford Lab), Matt Parker (Maryland Sea Grant), Beth Darrow and Jessica Kinsella (University of North Carolina Wilmington). Thanks to Christine Jacek and Cori Rose (US Army Corps of Engineers), Christian Petitpas (MA Division of Marine Fisheries), Russ Babb (NJ Department of Environmental Protection), Brian Thompson (CT Department of Energy and Environmental Protection), and David Carey (CT Bureau of Aquaculture) for discussions regarding regulatory needs and integration of the tool into shellfish aquaculture permitting.

## Author Contributions

**Conceptualization:** Julie M. Rose, Christopher Schillaci.

**Data curation:** Ryan Morse.

**Formal analysis:** Julie M. Rose, Ryan Morse.

**Funding acquisition:** Julie M. Rose, Christopher Schillaci.

**Investigation:** Julie M. Rose.

**Methodology:** Julie M. Rose, Ryan Morse.

**Project administration:** Julie M. Rose.

**Software:** Ryan Morse, Christopher Schillaci.

**Supervision:** Julie M. Rose.

**Validation:** Julie M. Rose.

**Visualization:** Ryan Morse.

**Writing – original draft:** Julie M. Rose, Ryan Morse, Christopher Schillaci.

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
