## [Decision Letter · Decision Letter 0]

23 Jul 2024

PONE-D-24-24623Development and application of an online tool to quantify nitrogen removal associated with harvest of cultivated eastern oystersPLOS ONE

Dear Dr. Rose,

Thank you for submitting your manuscript to PLOS ONE. After careful consideration, we feel that it has merit but does not fully meet PLOS ONE’s publication criteria as it currently stands. Therefore, we invite you to submit a revised version of the manuscript that addresses the points raised during the review process.

We look forward to receiving your revised manuscript.

Kind regards,

José A. Fernández Robledo, Ph.D.

Academic Editor

PLOS ONE

Journal Requirements:

3. Thank you for uploading your study's underlying data set. Unfortunately, the repository you have noted in your Data Availability statement does not qualify as an acceptable data repository according to PLOS's standards.

**Additional Editor Comments:**

Dear Dr. Rose.

the reviewers have submitted their reviews. Only minor comments. Please pay attention to address Review #2 "I went to the model and found it not so intuitive but I’m an old guy. I got the following error statements on two tries" to make sure the webpage does not give errors.

Regards,

-j

Reviewers' comments:

Reviewer's Responses to Questions

**Comments to the Author**

1. Is the manuscript technically sound, and do the data support the conclusions?

Reviewer #1: Yes

Reviewer #2: Yes

2. Has the statistical analysis been performed appropriately and rigorously? 

Reviewer #1: Yes

Reviewer #2: Yes

3. Have the authors made all data underlying the findings in their manuscript fully available?

Reviewer #1: Yes

Reviewer #2: Yes

4. Is the manuscript presented in an intelligible fashion and written in standard English?

Reviewer #1: Yes

Reviewer #2: Yes

5. Review Comments to the Author

**Reviewer #1:** Development and application of an online tool to quantify nitrogen removal associated

with harvest of cultivated eastern oysters

PONE-D-24-24623

As has been discussed in this well written and carefully documented manuscript, the use of shellfish aquaculture provides multiple social, economic and ecological benefits to communities. Among these benefits is the removal of excess nutrient, particularly nitrogen. There are few technologies which address in-water nutrient removal which partially explains the popularity of restorative aquaculture, and shellfish restoration. Coincidentally, commercial aquaculture of oysters has experienced a renaissance in the last decade along the Atlantic seaboard and NGO’s such as The Nature Conservancy are spearheading a global effort for shellfish restoration.

Rose et al. methodically examined the literature and worked with a wide network of shellfish scientists, and federal and state agencies for information regarding biological metrics for oysters grown in aquaculture farms from the states of Maine to Virginia. Using available data on the shell weight, shell size, and % nitrogen in the tissue and shell, the authors have constructed a statistically rigorous application to calculate nitrogen removal from an aquaculture farm which can be used during the aquaculture application process as a part of the list of community benefits. The easy-to-use computer tool will be a boon to those in the process of permitting an aquaculture site.

I believe this manuscript merits acceptance in Plos One. The database underpinning this much needed tool was carefully considered and expertly analyzed by the authors to determine the appropriate relationships between the oyster biological metrics and nitrogen concentrations upon which the equations were developed, and the tool was created. I believe the analyzes were appropriate to address the questions. The tables and figures were very clearly presented. My few suggestions follow.

With justified text it can be difficult to control spacing, but the authors should check spacing between sentences.

L 412 delete ‘a’

Table 4 add to caption ‘ploidy, cultivation practices, and regions’

L 511 I realize that one of the aims was to create an easy-to-use tool but I am concerned that the difference of nearly 100 lbs of N between triploidy and diploidy is considerable when scaled up. If this tool become widely accepted, is it possible to construct a set of equations for each ploidy?

L 597 ‘these data’

**Reviewer #2:** This ms describes a new synthesis of existing data in the form of development of an online model that predicts nitrogen removal by farmed oysters. If the new synthesis is “original research” then it is appropriate for PLOS ONE. All stats were sensible and interpreted appropriately. Figures 4 – 8 are pretty but most don’t seem necessary for model development? Maybe needed for other reasons? Perhaps move to “Supporting Information”?

It briefly sets the new model in the context of two similar models focused on other geographic areas. I would like to have seen some kind of comparative assessment.

I went to the model and found it not so intuitive but I’m an old guy. I got the following error statements on two tries:

“Warning: Position guide is perpendicular to the intended axis.

i Did you mean to specify a different guide `position`?

Warning: Removed 3 rows containing missing values or values outside the scale range

(`geom_bar()`).”

Bottom line is: this manuscript describes a valuable contribution (the model) and it should be published, but some revisions are needed.

6. PLOS authors have the option to publish the peer review history of their article (what does this mean?). If published, this will include your full peer review and any attached files.

Reviewer #1: No

Reviewer #2: No

---

## [Author Response · Author response to Decision Letter 0]

31 Jul 2024

Reviewer #1: Development and application of an online tool to quantify nitrogen removal associated

with harvest of cultivated eastern oysters

PONE-D-24-24623

As has been discussed in this well written and carefully documented manuscript, the use of shellfish aquaculture provides multiple social, economic and ecological benefits to communities. Among these benefits is the removal of excess nutrient, particularly nitrogen. There are few technologies which address in-water nutrient removal which partially explains the popularity of restorative aquaculture, and shellfish restoration. Coincidentally, commercial aquaculture of oysters has experienced a renaissance in the last decade along the Atlantic seaboard and NGO’s such as The Nature Conservancy are spearheading a global effort for shellfish restoration.

Rose et al. methodically examined the literature and worked with a wide network of shellfish scientists, and federal and state agencies for information regarding biological metrics for oysters grown in aquaculture farms from the states of Maine to Virginia. Using available data on the shell weight, shell size, and % nitrogen in the tissue and shell, the authors have constructed a statistically rigorous application to calculate nitrogen removal from an aquaculture farm which can be used during the aquaculture application process as a part of the list of community benefits. The easy-to-use computer tool will be a boon to those in the process of permitting an aquaculture site.

I believe this manuscript merits acceptance in Plos One. The database underpinning this much needed tool was carefully considered and expertly analyzed by the authors to determine the appropriate relationships between the oyster biological metrics and nitrogen concentrations upon which the equations were developed, and the tool was created. I believe the analyzes were appropriate to address the questions. The tables and figures were very clearly presented.

Thank you very much for your positive review!

 My few suggestions follow.

With justified text it can be difficult to control spacing, but the authors should check spacing between sentences.

We have searched the document for instances of double-space and removed two identified.

L 412 delete ‘a’

Done

Table 4 add to caption ‘ploidy, cultivation practices, and regions’

Done

L 511 I realize that one of the aims was to create an easy-to-use tool but I am concerned that the difference of nearly 100 lbs of N between triploidy and diploidy is considerable when scaled up. If this tool become widely accepted, is it possible to construct a set of equations for each ploidy?

One of the advantages of the online tool approach used here is that updates to the calculations are very straightforward when new data becomes available, and research is underway for future expansion of tool functions. Currently, the Chesapeake Bay is crediting triploids at higher nitrogen removal than diploids, based on their literature review from Maryland and Virginia only. This literature did not contain a direct comparison of diploids and triploids grown under identical conditions. A new paper published this year did that comparison, and found only small differences between diploids and triploids on two farms in Virginia and Maryland. These data are included in our nutrient tool and data repository. One additional unpublished dataset from Maine also suggests triploids are not removing nitrogen at higher rates than diploids, and these data are also included here. This is an evolving field in the literature, and until more data across broader geography is available, we feel that it is currently more appropriate to have a single equation for both diploids and triploids. We already have the structure in place in the tool to collect information on ploidy, and the equations can be easily updated.

L 597 ‘these data’

Done

Reviewer #2: This ms describes a new synthesis of existing data in the form of development of an online model that predicts nitrogen removal by farmed oysters. If the new synthesis is “original research” then it is appropriate for PLOS ONE. All stats were sensible and interpreted appropriately. Figures 4 – 8 are pretty but most don’t seem necessary for model development? Maybe needed for other reasons? Perhaps move to “Supporting Information”?

We appreciate the suggestion to streamline the main body of the manuscript. We feel that the figures in question illustrate design choices to simplify the model, and are important visual support for using a single equation across ploidy, cultivation practice, and region. Our preference would be to retain these figures in the main body of the manuscript instead of moving them to Supporting Information.

It briefly sets the new model in the context of two similar models focused on other geographic areas. I would like to have seen some kind of comparative assessment.

Thank you very much for this suggestion, we conducted the assessment recommended and have incorporated new text into the Discussion section as follows:

“We compared our results to the Florida Shellfish Farm Nitrogen Calculator using inputs of 1,000,000 oysters harvested at a shell height of 3 inches. The Florida Calculator predicted nitrogen removal of 265 pounds, and the ANRC tool predicted 311.6 pounds of nitrogen. We also compared our tissue-only nitrogen removal results to the Chesapeake Bay diploid regression. The Chesapeake Bay diploid tissue regression predicted nitrogen removal of 198 pounds, and the ANRC tool predicted 194.5 pounds.”

I went to the model and found it not so intuitive but I’m an old guy. I got the following error statements on two tries:

“Warning: Position guide is perpendicular to the intended axis.

i Did you mean to specify a different guide `position`?

Warning: Removed 3 rows containing missing values or values outside the scale range

(`geom_bar()`).”

We apologize for the error messages! We believe that this error was due to an update to the Posit Connect server, which has since been resolved, and the tool should now be functioning properly. We are unable to replicate any of these errors in the current version of the tool across multiple browsers and on two separate mobile platforms. We plan to roll out the tool to a broad user group of shellfish growers and other industry members in the fall, which should allow us to conduct thorough testing across many platforms, and make updates to user interface based on real-world user feedback.

Bottom line is: this manuscript describes a valuable contribution (the model) and it should be published, but some revisions are needed.

Thank you very much for your helpful comments!

---

## [Decision Letter · Decision Letter 1]

21 Aug 2024

Development and application of an online tool to quantify nitrogen removal associated with harvest of cultivated eastern oysters

PONE-D-24-24623R1

Dear Dr. Rose,

We’re pleased to inform you that your manuscript has been judged scientifically suitable for publication and will be formally accepted for publication once it meets all outstanding technical requirements.

Kind regards,

José A. Fernández Robledo, Ph.D.

Academic Editor

PLOS ONE

Additional Editor Comments (optional):

Dear Dr. Rose,

Both reviewers agree that after the latest edits, you have answered their comments.

I am looking forward to see how the scientific community and public in general takes advantage of this new tool.

Sincerely,

-j

Reviewers' comments:

Reviewer's Responses to Questions

**Comments to the Author**

1. If the authors have adequately addressed your comments raised in a previous round of review and you feel that this manuscript is now acceptable for publication, you may indicate that here to bypass the “Comments to the Author” section, enter your conflict of interest statement in the “Confidential to Editor” section, and submit your "Accept" recommendation.

Reviewer #1: All comments have been addressed

Reviewer #2: All comments have been addressed

2. Is the manuscript technically sound, and do the data support the conclusions?

Reviewer #1: (No Response)

Reviewer #2: Yes

3. Has the statistical analysis been performed appropriately and rigorously? 

Reviewer #1: (No Response)

Reviewer #2: Yes

4. Have the authors made all data underlying the findings in their manuscript fully available?

Reviewer #1: (No Response)

Reviewer #2: Yes

5. Is the manuscript presented in an intelligible fashion and written in standard English?

Reviewer #1: (No Response)

Reviewer #2: Yes

6. Review Comments to the Author

Reviewer #1: (No Response)

Reviewer #2: All revisions are acceptable to me. The manuscript is now acceptable, and their attention to the model itself based on my comments were good.

7. PLOS authors have the option to publish the peer review history of their article (what does this mean?). If published, this will include your full peer review and any attached files.

Reviewer #1: No

Reviewer #2: No

---

## [Editor Report · Acceptance letter]

27 Aug 2024

PONE-D-24-24623R1 

PLOS ONE

Dear Dr. Rose, 

I'm pleased to inform you that your manuscript has been deemed suitable for publication in PLOS ONE. Congratulations! Your manuscript is now being handed over to our production team.

Kind regards, 

on behalf of

Dr. José A. Fernández Robledo 

Academic Editor

PLOS ONE